# Hitting two BSM particles with one lepton-jet: Search for a top partner decaying to a dark photon, resulting in a lepton-jet

Karien du Plessis[1], M. M. Flores[1,2], Deepak Kar[1⋆],
Sukanya Sinha[1] and Hannah van der Schyf[1]

**1** School of Physics, University of Witwatersrand, Johannesburg, South Africa
**2** National Institute of Physics, University of the Philippines,
Diliman, Quezon City, Philippines

⋆ deepak.kar@cern.ch

## Abstract

A maverick top partner model, decaying to a dark photon was suggested in Ref. [1]. The dark photon decays to two overlapping electrons for dark photon masses of $\approx$ 100 MeV, and results in a so-called lepton-jet. The event includes a top quark as well, which results in events with two boosted objects, one heavy and the other light. We propose a search strategy exploiting the unique signal topology. We show that for a set of kinematic selections, both in hadronic and leptonic decay channel of the SM top quark, almost all background can be eliminated, leaving enough signal events up to top partner mass of about 3.5 TeV for the search to be viable at the LHC Run 2. Further, integrated luminosity needed for the proposed search to be sensitive is presented as a function of top partner mass.



# 1 Introduction

The Standard Model (SM) of particle physics has by far been successful in explaining almost all the experimentally observed microscopic phenomena. However, it is deemed an incomplete theory due to a number of reasons, one of which is its failure to explain dark matter (DM), whose existence is empirically established from various astrophysical data [2, 3]. However, little is known regarding its interaction with itself and with the elementary particles of SM. One possibility is that dark matter particles may interact through some dark force akin to the electromagnetic force felt by ordinary particles. This leads to the conjecture of the existence of a new gauge boson that would mediate this dark force, with the leading interactions in the small mixing limit being the QED current, analogous to the role of the photon in Quantum Electrodynamics. For this reason, the new gauge boson is referred to in literature as the *dark photon* [4–8] (which will be denoted in this paper as $\gamma_d$), although the terms *paraphoton* [9] and *U-boson* [10] have been used in the past.

In this paper, we propose a new search strategy for a chimera-like scenario, where a dark photon does not arise from known SM particles, but rather from a hypothetical vector-like quark, dubbed the *maverick top partner (abbreviated as VLT)* [1] [11–14] with unconventional decays. The primary model is motivated by [1], in which a top-partner is charged under both the SM and a new gauge $U(1)_d$ group, where the SM is neutral under the $U(1)_d$.

The paper is divided into the following sections: in Section 2 we discuss the general idea of maverick top partners and the model in concern, followed by event generation procedure which is provided in 2.2. In Section 3, we discuss the particle - level analysis performed for the model in question, and estimate our ability to experimentally probe such a chimera-like scenario, and finally present our conclusions in Section 4.

# 2 Maverick Top Partners

## 2.1 Brief Survey of the Model

There are theoretical reasons to believe that dark photons could be massless or massive [8], and their respective features and experimental signatures are quite distinct. In this paper we focus on the massive dark photon since it is more readily accessible in the collider searches (due to the fact that it couples to ordinary matter).

Typical production mechanisms of massive dark photons include [8, 15]:

- Bremsstrahlung, whereby an incoming electron scatters off a nuclei target (Z) and emits dark photon, i.e., $e^- Z \to e^- Z \gamma_d$ [16];

- Annihilation, whereby an electron-positron pair annihilates into a photon and a dark photon, i.e., $e^- e^+ \to \gamma \gamma_d$ [17]; and

- Drell-Yan, whereby a quark-antiquark pair annihilates into a dark photon, which consequently decays into a lepton pair or hadrons, i.e., $q\bar{q} \to \gamma_d \to l^- l^+$ or $h^- h^+$ [18].

Here we consider a special production mechanism where the dark photon comes from a VLT, and thus the production chain will look like VLT$\to t\gamma_d$, where $t$ is the SM top quark, as shown in the bottom diagrams of Fig 1.

The decay modes of the massive dark photon is further divided into whether it is visible or invisible, with the borderline requirement of visibility being $m_{\gamma_d} > 2m_e \simeq 1$ MeV because then it can decay into SM charged particles (electron-positron pairs being the extreme case) which leave a signature in the detectors. Otherwise, it cannot decay into known SM charged

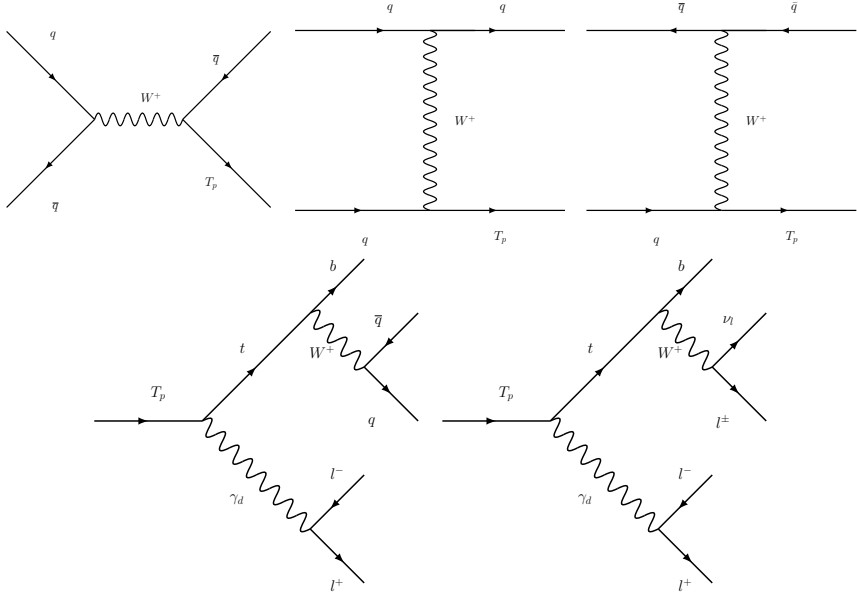

Figure 1: Feynman diagrams showing the production modes of the VLT, denoted by $T_p$ (top row) and considered decay modes of VLT (bottom row) going to a top quark and a dark photon. The conjugate processes for the $t$-channel production modes are not shown here. The dark photon, denoted by $\gamma_d$, decays to two oppositely charged same flavour leptons. The top quark can decay hadronically (bottom left) or leptonically (bottom right), which will be considered separately.

particles and the decay is therefore invisible. We choose to study this extreme case of dark photon decaying only into electron-positron pairs since these final states result into the lesser studied unusual topology known as *lepton-jets* [19–21]. This choice constrains our massive photon to have a mass between 1 MeV to 200 MeV, otherwise the branching ratio (BR) into electron-positron pairs is not 100% and we get non-zero BR into muon-antimuon pairs [1,15].

Hence, we perform a phenomenological study on final states involving a lepton-jet in association with the top quark, investigating both the scenarios where the top quark decays hadronically or leptonically.

No evidence in favour of VLTs has been found at the Large Hadron Collider (LHC) when probing via its traditional decays into SM particles. Currently, the most stringent limits on VLT masses are set by the ATLAS and CMS experiments [22–25]. The excluded masses for VLT depend on the branching ratios (BR) assumed; and for extreme values of 100% BR, a singlet VLT is excluded for masses below 2 TeV. Hence, nontraditional decays are searched for, including decay into the dark photon which becomes dominant provided that its mass is very less compared to the SM electroweak sector ($m_{\gamma_d} \ll m_Z$) [1] [1]. This appealing scenario provides a probe for light dark sector by searching for heavy particles at the LHC, thus bridging the two heavy and light mass regimes of BSM particles, VLTs and dark photons respectively.

In order for the VLT decays into the dark photon to be dominant, we follow the model discussed in-depth in [1], where a new Abelian gauge symmetry $U(1)_d$ is introduced whose gauge boson is the dark photon itself (i.e., SM is extended to $SU(3) \times SU(2)_L \times U(1)_Y \times U(1)_d$). The SM particles are singlets under this new symmetry while the VLT has a charge +1. Specifically, this VLT is a singlet under $SU(2)_L$ with SM hypercharge $Y = 2/3$. The details of the

---

[1]More accurately, the enhancement of the VLT decay into dark photons is due to the vev ratio $(v_{EW}/v_d)^2$ where $v_{EW}$ is the Higgs boson vacuum expectation value (vev) and $v_d$ is the vev of the dark sector Higgs boson. Since there is a quadratic dependence on the vevs, the mass gap between $m_{\gamma_d}$ and $m_Z$ does not have to be very large for the decays to be dominant, though the case where $m_{\gamma_d} \ll m_Z$ only serves to enhance this predominant decay.

Table 1: Field content of the maverick top partner model and their corresponding charges. The usual third generation SM quarks are denoted by $t_{1R}$, $b_R$, and $Q$ while $\Phi$ is the SM Higgs doublet. In addition, $t_2$ is the $SU(2)_L$ singlet VLT, and $H_d$ is the $U(1)_d$ Higgs field. $Y$ is the usual SM hypercharge while $Y_d$ is the charge of the new $U(1)_d$ gauge symmetry.

|  | $SU(3)$ | $SU(2)_L$ | $Y$ | $Y_d$ |
|---|---|---|---|---|
| $t_{1R}$ | 3 | 1 | 2/3 | 0 |
| $b_R$ | 3 | 1 | -1/3 | 0 |
| $Q_L = \begin{pmatrix} t_{1L} \\ b_L \end{pmatrix}$ | 3 | 2 | 1/6 | 0 |
| $\Phi$ | 1 | 2 | 1/2 | 0 |
| $t_{2L}$ | 3 | 1 | 2/3 | 1 |
| $t_{2R}$ | 3 | 1 | 2/3 | 1 |
| $H_d$ | 1 | 1 | 0 | 1 |

field content and their corresponding charges are given in Table 1. Again, we defer the discussion of the full model with the relevant expressions for the Lagrangians and interactions in [1] where it was fully worked out. We simply state that in addition to the usual SM parameters, the new external free parameters that are added are: the VLT mass $M_{VLT}$, mass of the dark photon $m_{\gamma_d}$, the dark sector Higgs boson vev $v_d$, the dark kinetic mixing parameter $\varepsilon$, the top-VLT mixing angle $\sin\theta_l^t$, the scalar mixing angle between the Higgs boson and dark Higgs boson $\sin\theta_S$, and the dark Higgs mass $m_{H_d}$. In this model, the dark photon gauge boson kinetically mixes with the SM $U(1)_Y$ field. While this opens up a decay channel of VLT into the dark photon (plus top), its decay into fully SM final states (such as $W/Z/h$ plus top) is still feasible. To enhance the former, we take the $U(1)_d$ scale to be significantly smaller than the electroweak scale ($m_{\gamma_d} \ll m_Z$) since the partial widths of VLT decay into $\gamma_d$ relative to SM electroweak bosons is proportional to $(v_{EW}/v_d)^2$, where $v_{EW} = 246$ GeV is the Higgs boson vacuum expectation value (vev) and $v_d$ is the vev of the dark sector Higgs boson.

It should be noted that VLTs at the LHC can either be pair produced ($pp \to T\bar{T}$), or produced singly in association with a jet ($pp \to T/\bar{T} + \text{jet}$). The single production mode is relatively model-dependent and depends on the mixing of the VLT with the SM quarks, whereas the pair production mode is more model-independent, depending on the colour representation, mass, and spin of the VLT. Thus, if the mixing angle, $\sin\theta_l^t$, of the VLT with the SM top quark is considerably small, pair production rates surpass those of single production [12]. However, by considering $\sin\theta_l^t \backsim 0.1$, the single production mechanism surpasses the pair production in a wide range of VLT masses, with the latter only outmatching the former in relatively low VLT masses ($M_{VLT} \lesssim 1000$ GeV) [1,12,26]. In this work, we only focus on the single VLT production since not only is it the more favored production phase space, the final state it produces is also relatively "cleaner" as opposed to pair production.

Enforcing that the kinetic mixing $\varepsilon$ is small in addition to the $m_{\gamma_d} \ll m_Z$ mass requirement, enhances VLT decay into dark photons, and the $U(1)_d$ gauge boson inherits couplings to SM particles through the electromagnetic current with coupling strength $\varepsilon e Q$ [1]. This in turn, kinematically opens up dark photon decays into $e^+e^-$, for the values of $m_{\gamma_d}$ considered in this paper.

## 2.2 Event Generation

Signal events were generated using MADGRAPH5 interfaced with PYTHIA8 [27] (using the default NN23LO1 parton distribution function set [28]) from UFO [29] files based on the mav-

erick top partner model in [1]. As outlined in the aforementioned reference, the free external parameters of the model are as follows (with their corresponding values): the dark sector Higgs boson vev $v_d = 10$ GeV, the dark kinetic mixing parameter $\varepsilon = 0.001$, the top-VLT mixing angle $\sin\theta_l^t = 0.1$, the scalar mixing angle between the Higgs and dark Higgs boson, $\sin\theta_S = 0.1$, and the dark Higgs boson mass $m_{H_d} = 10$ GeV. The other two free parameters are the $M_{VLT}$ and $m_{\gamma_d}$ masses. We vary the $M_{VLT}$ from 1 TeV to 5 TeV (with cross sections of 47.6 fb to 0.0054 fb respectively) to make sure that we remain at the phase space where it is singly produced. Also, we chose $m_{\gamma_d} = 0.1$ GeV which ensures that the branching ratio (BR) to $e^+e^-$ is 1, as can be seen in Fig. 8(b) of Ref. [1].

For each signal mass point, 100,000 events were generated, while for each background processes, 1 million events were generated. All the background processes have been generated using PYTHIA8 as well. While a leading order generator cannot model all the background accurately, our aim here is to get the general characteristics of the background processes, and suggest ways to reduce them, for which PYTHIA8 is adequate. In any case, the region of interest is the tail of reconstructed VLT mass distribution, so only general features can be studied in a particle level analysis with limited Monte Carlo statistics. Finally, all distributions below are normalised using an integrated luminosity of 300 fb$^{-1}$, corresponding to projected combined integrated luminosity combining LHC Run 2 and Run 3.

# 3 Analysis strategy

## 3.1 Overview

The signal is characterised by an opposite sign electron pair with high $p_T$ and very little angular separation. They form what is known as a lepton-jet (more specifically electron-jet in our case), where tracks from the two electrons will overlap, and the energy deposits will be collected in a single jet. Electron reconstruction algorithm used in LHC experiments will mostly mis-reconstruct it as a single electron. So unlike standard searches, electron multiplicity is a misleading observable here.

Lepton-jets have been studied sparingly at the LHC [30–32], mostly to search for SUSY and Higgs portal models with relatively high mass bosons compared to the dark photon considered here. As mass calibration for jets in experiments is deemed unreliable below 10 GeV, we are in no position to exploit the lepton-jet mass. However, since the lepton-jet will be boosted, the mass over transverse momentum ratio (subsequently referred to as $m/p_T$) of such jets can be used as a discriminating variable, even accounting for the inherent uncertainty in the measured mass value. The *standard* jets from quarks and gluons tend to have a much higher mass compared to hundreds of MeV in this signal. Experimentally, the lepton-jet is expected to have a larger than usual fraction of energy deposited in electromagnetic calorimeter, which can be exploited as well.

The event also contains a top quark, and at least one more jet coming from the initial quark. A sketch of the event topology is shown in Fig. 2. We do note this is aid for visualisation, not necessarily an accurate indicator of the event kinematics.

The decay mode of the top quark determines the full event topology. We investigate the two scenarios of the top quark decaying hadronically and leptonically separately.

We note that due to the lepton-jet being very boosted, the resulting two electrons will have a very skewed energy (hence transverse momentum) distribution, with one usually produced with a much higher $p_T$. This has crucial implications in the leptonic channel, as we will see. We also note that the initial quark produced along with the VLT can be a bottom quark, so we expect two b-tagged jets in some events.

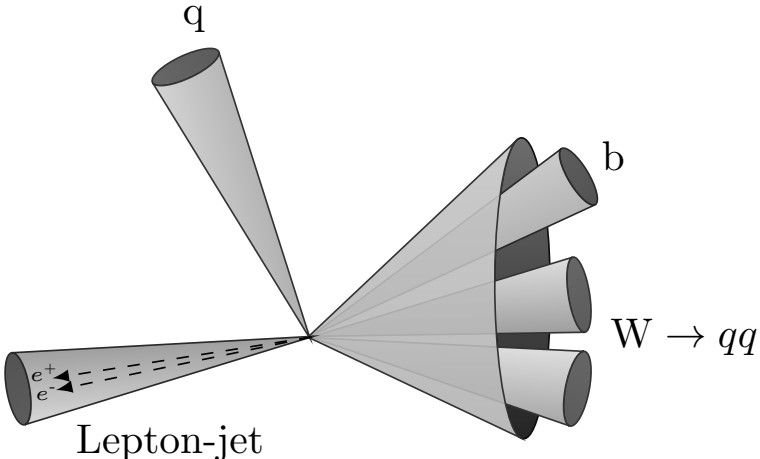

Figure 2: A cartoon of different final state configuration in a typical signal event, for the hadronic decay mode of the top quark. The large-radius jet (J10) on the right originates from the top quark, which includes three jets (J4), one of them is b-tagged. One of the other two jets (J4) is the lepton-jet candidate, which includes the oppositely charged electron pair almost overlapping with each other. In the leptonic decay mode of the top quark, it will contain a charged lepton and corresponding neutrino, along with the b-tagged jet.

In the following analysis, we will use objects as they are commonly used in experiments. Jets will be the ones reconstructed using anti-$k_t$ [33] algorithm with radius parameter of 0.4, $p_T > 30$ GeV and $|\eta| < 4.4$. We will refer them to as J4 subsequently for clarity. We note that the lepton-jet candidate is a J4. Large radius jets are reconstructed using anti-$k_t$ algorithm with radius parameter of 1.0, with $p_T > 150$ GeV and $|\eta| < 2$. We have not used any grooming as it is a particle level study without any pileup, but application of standard trimming or soft-drop would not change any conclusions. We will refer them to as J10 subsequently for clarity. We note that the boosted top quark candidate is a J10. Muons or neutrinos are not used as jet inputs, and the presence of a b-hadron inside a jet is used to determine if the jet is b-tagged or not. Lepton are chosen with $p_T > 25$ GeV and $|\eta| < 2.5$, dressed with photons within a radius of 0.1 and not originating from hadron decays. The missing transverse momentum is calculated from the negative four-vector sum of all visible particles. The most inclusive suggested trigger will be a single electron trigger with appropriate threshold. Rivet analysis toolkit [34] has been used.

### 3.2 Analysis with the top quark decaying hadronically

The signal topology in hadronic channel is two electrons being identified as one as a part of a high $p_T$, low mass lepton-jet, a boosted top quark resulting in a top-jet, and at least another jet. The event selection requirement is at least one electron, no muons, at last one J10, and at least three J4s, with at least one of them being b-tagged. One of the J4s will need to be the lepon-jet candidate, as discussed below.

The largest background by far is the usual multijet processes, where a jet is misreconstructed as an electron. Top quark pair production and all-hadronic and (electron-channel) semileptonic decay modes will consist of the other significant background processes. The cross-sections of other possible background processes, like associated production of $Z/W/\gamma$ with a top quark and additional jets are tiny, and therefore neglected.

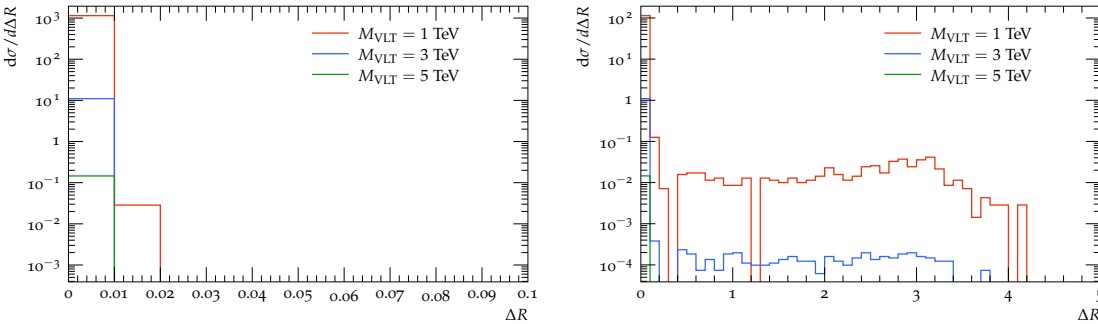

Figure 3: Distributions of $\Delta R$ between the two electrons from the dark photon decay (left) and $\Delta R$ between leading electron and closest J4 (right) for three representative signal points, corresponding to VLT masses of 1, 3 and 5 TeV are shown for the hadronic channel analysis.

The lepton-jet candidate is the J4 closest in $\Delta R$ to the leading $p_T$ electron. This is verified by checking the $\Delta R$ between the two electrons and between the leading electron and the closest J4 in Fig. 3.

As expected, the two electrons are almost on top of each other, and there is always a J4 collinear with them. In order to be consistent with the experimental signature, we require minimum of only one electron, not two. However, for multijet and hadronic $t\bar{t}$ backgrounds, this one electron requirement has to be relaxed, as jets misidentified as electrons will result in these reducible backgrounds. Here we have scaled the cross-sections by 0.1 to mimic the electron misidentification rate, which is a conservative estimate derived from [35]. Requiring a *tighter* electron identification criteria will help in reducing these backgrounds.

The top-jet candidate is a highest mass J10, satisfying a top-mass window (chosen rather liberally to be between 125–225 GeV) requirement. The J10 is also required to contain a b-tagged J4 within $\Delta R < 1$. While any of the standard top-tagging techniques [36, 37] or requirements on jet substructure observables can be used to increase the purity of the top-jet selection, we leave that for the experimental analysis.

Since there is no real electron in multijet and hadronic $t\bar{t}$ background events, we have considered the J4 farthest from the top-jet candidate as the lepton-jet. Then the invariant mass of the top partner is reconstructed from four-momenta of the lepton-jet and top-jet candidates. This is the key observable for the search.

Kinematic selections are used based on signal characteristics to reduce the background contributions. Fig. 4 shows lepton-jet and top-jet $p_T$ distributions, and based on these, a 200 GeV requirement is applied on both. Additionally based on Fig. 5, a $\Delta\phi > 1.5$ requirement is applied between the lepton-jet and top-jet candidate, as they are expected to be more back-to-back than most background processes. After all the above kinematic requirements, we are still left with a significant background, as can be seen in the reconstructed VLT mass distribution on the right of Fig. 5.

Then in Fig. 6, $m/p_T$ for signals and background processes are compared, and requiring the ratio to be less than 0.01 essentially keeps only the signal, as can be seen in the right figure.

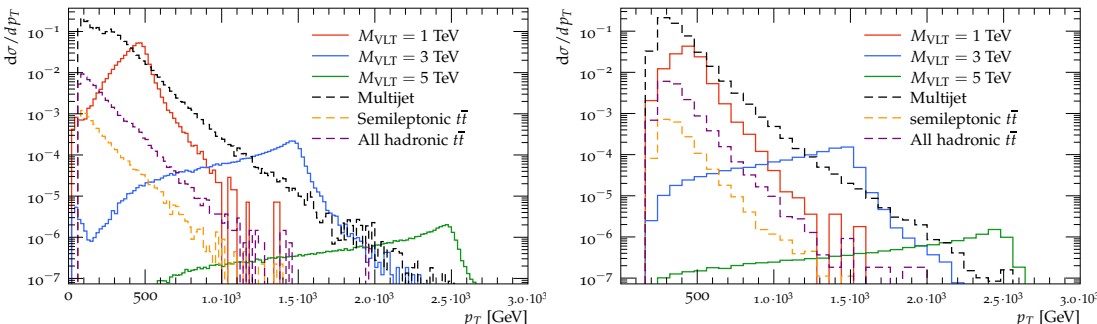

Figure 4: Distributions of lepton-jet candidate $p_\mathrm{T}$ (left) and the top-jet candidate $p_\mathrm{T}$ (right) for three representative signal points, corresponding to VLT masses of 1, 3 and 5 TeV and dominant background processes are shown, for the hadronic channel analysis.

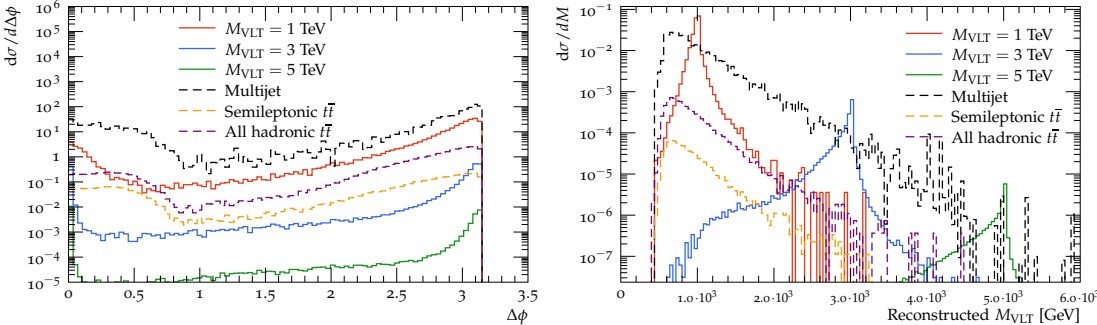

Figure 5: Distributions of $\Delta\phi$ between the lepton-jet and hadronic top-jet candidates and reconstructed VLT mass after all kinematic requirements for three representative signal points and dominant background processes are shown, for the hadronic channel analysis.

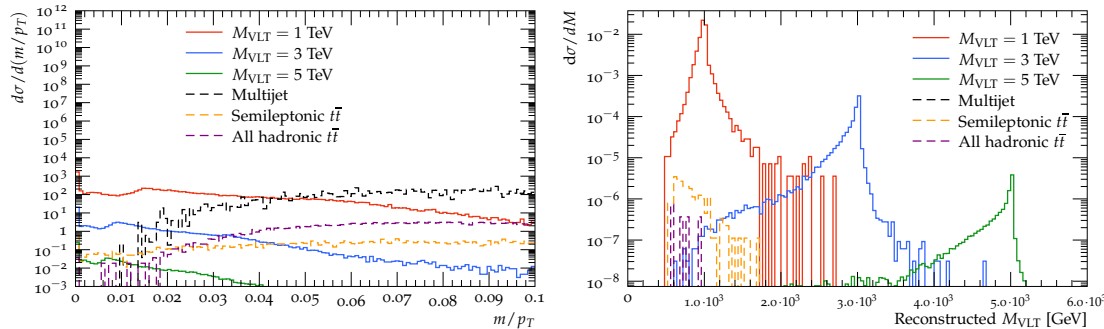

Figure 6: Distributions of lepton-jet candidate $m/p_T$ ratio before (left) and reconstructed VLT mass after the $m/p_T < 0.01$ requirement (right) for three representative signal points, corresponding to VLT masses of 1, 3 and 5 TeV and dominant background processes are shown, for the hadronic channel analysis.

Table 2: Effect of kinematic requirements on hadronic channel signal efficiencies, corresponding to VLT masses of 1, 3 and 5 TeV. The numbers represent the percentage of events remaining after each requirement. The individual requirements are mentioned in the text.

| Requirement | Signal efficiency reduction for VLT mass in %: | | |
|---|---|---|---|
|  | 1 TeV | 3 TeV | 5 TeV |
| Lepton multiplicity | 97 | 98 | 98 |
| J4 and J10 multiplicity | 93 | 94 | 92 |
| Top mass window | 75 | 79 | 71 |
| B-jet multiplicity | 68 | 75 | 68 |
| lepton-jet $p_{\text{T}}$ | 64 | 73 | 67 |
| Top-jet $p_{\text{T}}$ | 64 | 73 | 67 |
| B-jet containment | 57 | 57 | 41 |
| Lepton-jet and top-jet separation | 56 | 56 | 40 |
| Lepton-jet $m/p_{\text{T}}$ | 17 | 29 | 27 |

A detailed study on the effect of these kinematic requirement on signal and background was performed and summarised in Table 2 for the signal points.

While the requirements enforced decrease the signal significantly, the $m/p_{\text{T}}$ requirement essentially makes it a zero background search. It is interesting to note that the signal efficiency is better for intermediate VLT masses compared to higher VLT masses, even though for latter case the $m/p_{\text{T}}$ requirement is less inefficient, as the jets are more boosted. However, more boosted jets also mean the requirement of $\Delta\phi > 1.5$ between the lepton-jet and top-jet candidate is less efficient for higher VLT masses. After all selections, for an integrated luminosity of 300 fb$^{-1}$, 2380 signal events remain for VLT mass of 1 TeV, and 35 events for VLT mass of 3 TeV, while the less than 1 event for VLT mass of 5 TeV makes that signal inaccessible. A detailed sensitivity study is performed in Section 3.4.

### 3.3 Analysis with the top quark decaying leptonically

The signal topology in leptonic channel is two electrons being identified as one as a part of a high $p_{\text{T}}$, a low mass lepton-jet, a top quark decaying leptonically, and at least another jet. Depending on lepton flavour from top quark decay, two scenarios are possible, termed electron and muon channels. At least two J4s, with at least one of them b-tagged is required. No requirement on J10 is used in leptonic channels.

In both cases, the top quark is reconstructed by using the pseudo-top method [38]. The algorithm starts with the reconstruction of the neutrino four-momentum. While the $x$ and $y$ components of the neutrino momentum are set to the corresponding components of the missing transverse momentum, the $z$ component is calculated by imposing the $W$ boson mass constraint on the invariant mass of the charged-lepton-neutrino system. If the resulting quadratic equation has two real solutions, the one with the smaller value of $|p_z|$ is chosen. If the discriminant is negative, only the real part is considered. The leptonically decaying $W$ boson is reconstructed from the charged lepton and the neutrino. The leptonic top quark is reconstructed from the leptonic $W$ and the b-tagged J4 closest in $\Delta R$ to the charged lepton. We will refer it to as the leptonic top candidate.

In the muon channel, the muon is always used to reconstruct the leptonic top. In the electron channel, a possible ambiguity can arise, as there are three electrons, two of them are expected to be overlapping, and the third from the top quark. We cannot *a priori* assume that the leading electron is from the dark photon decay, or the electron closest to the b-tagged

jet is from the top quark decay, as a considerable fraction of events have two b-tagged jets, the second one from the initial quark. However, we have found that the electron pair with the highest $p_T$ comes from the dark photon about 99.5% of the time, so we use this pair as the single merged electron seeding the lepton-jet, as seen by the detector. The remaining electron is used for leptonic top reconstruction. Then the invariant mass of the top partner is reconstructed from four-momenta of the lepton-jet and leptonic top candidates.

As in the hadronically decaying top quark scenario above, we need to be careful about lepton multiplicity when considering the background processes. For the muon channel, in order to be consistent with the experimental signature, we require at least one electron and at least one muon. The largest background is the dileptonic mixed flavour $t\bar{t}$ process, which is an irreducible background. Since mis-reconstruction of muons as jets are relatively rare as compared to electrons, we do not have to worry about background that does not contain a real muon. The semileptonic $t\bar{t}$ process with a muon will be a reducible background, where a J4 from leptonic top can mimic the lepton-jet. As before, for this case, we have loosened the electron requirement, and applied a normalisation factor of 0.1.

For the electron channel, we start by requiring at least three electrons, and no muons. However, then the merged electron is used, so the effective requirement is two electrons. The largest background is the dileptonic $t\bar{t}$ process with both top quarks decaying to electrons, which is an irreducible background. The semileptonic $t\bar{t}$ process with an electron will be a reducible background. where a J4 from hadronic top can mimic the lepton-jet or the leptonic top. As before, for this case, we have loosened the electron requirement, and applied an extra normalisation factor of 0.1.

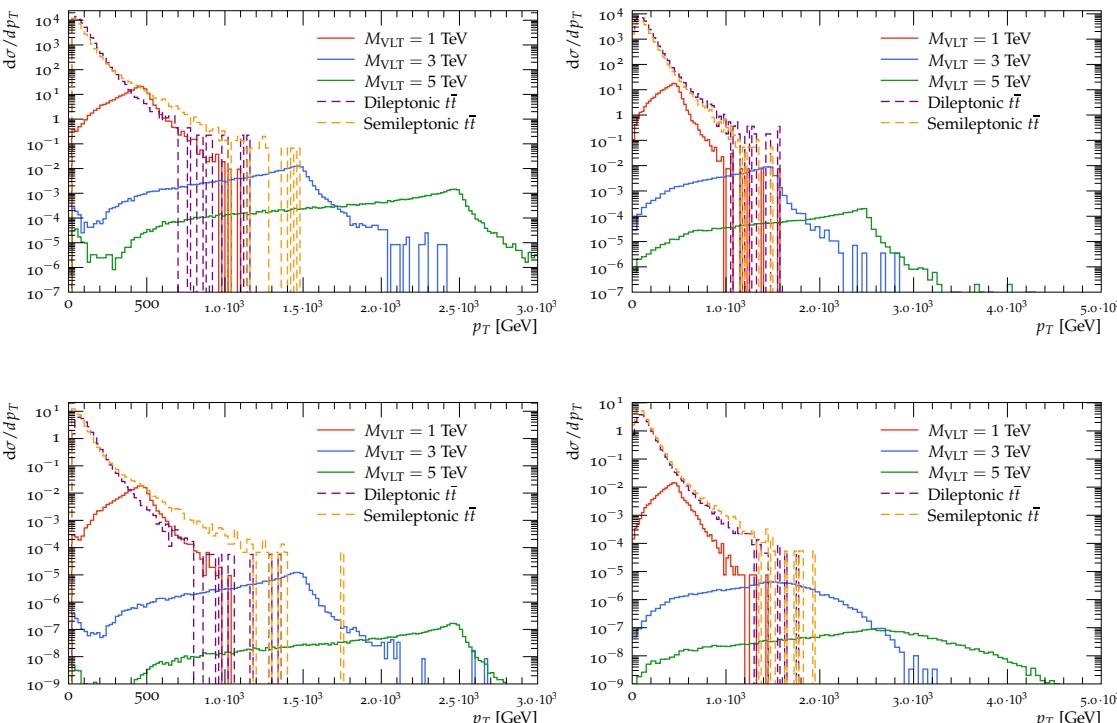

Figure 7: Distributions of lepton-jet candidate $p_T$ (top row) and the leptonic top candidate $p_T$ (bottom row) for muon (left) and electron (right) channels for three representative signal point, corresponding to VLT masses of 1, 3 and 5 TeV and dominant background processes are shown.

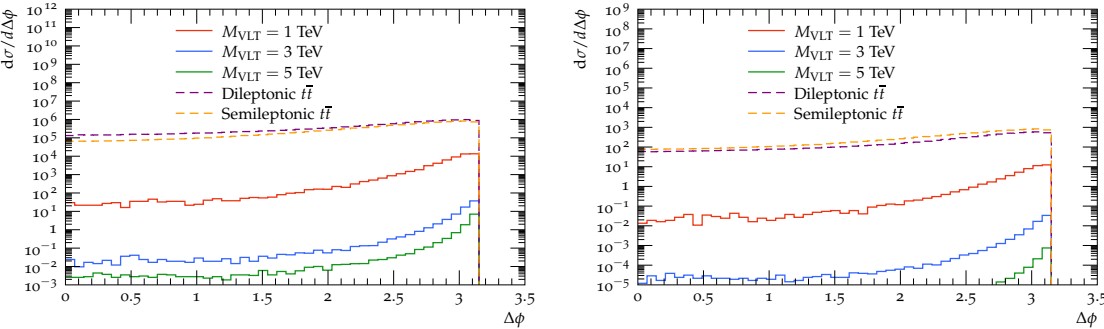

Figure 8: Distributions of $\Delta\phi$ between the lepton-jet candidate and the leptonic top candidate for the muon channel (left) and the electron channel (right) for three representative signal points, corresponding to VLT masses of 1, 3 and 5 TeV and dominant background processes are shown.

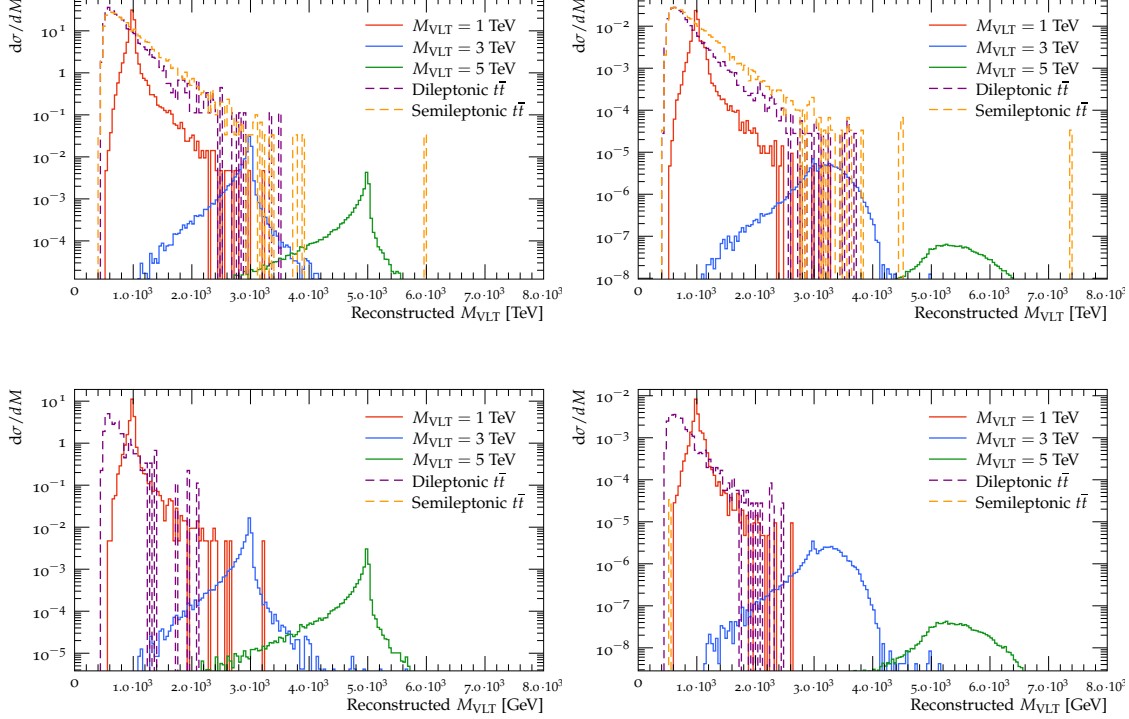

Figure 9: Distributions of reconstructed VLT mass before (top row) and after the $m/p_T < 0.01$ requirement, for muon (left) and electron (right) channels for three representative signal points, corresponding to VLT masses of 1, 3 and 5 TeV and dominant background processes are shown.

The other significant backgrounds can be $W/Z$ boson (decaying to electrons) with jets. As the $Z$-boson mass is much higher than the dark photon mass, the electrons rarely end up overlapping, and the $b$-jet requirement essentially gets rid of almost all the $W$+jets contribution. We have not explicitly considered $\tau$ decay modes of the top quark as the hadronic decay mode will be equivalent to the fully hadronic signature considered, and the leptonic decay modes will be similar to the electron and muon channels.

Table 3: Effect of kinematic requirements on leptonic channel signal efficiencies, corresponding to VLT masses of 1, 3 and 5 TeV. The numbers represent the percentage of events remaining after each requirement. The individual requirements are mentioned in the text.

| Requirement | Signal efficiency reduction for VLT mass in %: | | | | | |
| | 1 TeV | | 3 TeV | | 5 TeV | |
| | El | Mu | El | Mu | El | Mu |
|---|---|---|---|---|---|---|
| Lepton multiplicity | 70 | 82 | 88 | 92 | 92 | 94 |
| J4 multiplicity | 62 | 72 | 84 | 88 | 88 | 92 |
| lepton-jet $p_T$ | 60 | 68 | 82 | 88 | 88 | 92 |
| Top jet $p_T$ | 58 | 66 | 82 | 88 | 88 | 92 |
| Top mass minimum | 58 | 64 | 82 | 86 | 88 | 92 |
| B-jet containment | 54 | 60 | 80 | 84 | 86 | 90 |
| Lepton and top separation | 52 | 58 | 78 | 82 | 84 | 88 |
| Lepton-jet $m/p_T$ | 16 | 18 | 38 | 40 | 54 | 56 |

The kinematic requirements are consistent with those applied in hadronic case, with requirements of lepton-jet $p_T$ and leptonic top candidate $p_T$ of 200 GeV. Most kinematic distributions are similar to the hadronic channel ones shown above, so we did not repeat them. The distribution of them is shown in Fig. 7.

Similarly the b-tagged jet is required to be with $\Delta R < 1$ of the leptonic top candidate, and a $\Delta \phi > 2.5$ requirement is applied between the lepton-jet and leptonic top candidate, the latter distribution is shown in Fig. 8. A loose minimum mass requirement of 100 GeV is applied for the leptonic top candidate. Finally the same $m/p_T$ requirement as before is applied, and it again helps to massively reduce all backgrounds, as can be seen in Fig. 9.

A detailed study on the effect of these kinematic requirement on signal and background was performed and summarised in Table 3 separately for electron and muon channels, as defined above. The initial drop is due to one of the leptons falling below the kinematic thresholds, and while the $m/p_T$ requirement does reduce the efficiencies drastically, it also eliminates almost all of the background, as seen above. We can also consider a requirement on missing transverse momentum, but this is always better to leave for a detector level analysis. Similarly, a requirement of about 100 GeV on the leading electron $p_T$ keeps almost all of the signal and will help in reducing backgrounds, but this is better optimised using the potential misreconstructed electrons at the detector level. After all selections and combining both channels, for an integrated luminosity of 300 fb$^{-1}$, 1456 signal events remain for VLT mass of 1 TeV, and 29 events for VLT mass of 3 TeV, while the less than 1 event for VLT mass of 5 TeV makes that signal inaccessible. A detailed sensitivity study is performed in Section 3.4.

## 3.4 Projected sensitivity

Although this is probably an idealised assumption in a particle level study, where detector effects and mis-measurements will result in a non-zero background, we can estimate the integrated luminosity needed as a function of VLT mass to exclude the signal if we observe no events, but expect at least three signal events [39, 40]. This is shown separately for hadronic and leptonic channels in Fig. 10, where we can see Run 2 data may already be sufficient to have sensitivity up to VLT mass of 3.5 TeV. The hadronic channel can be seen to have larger sensitivity at lower VLT masses, while at higher masses the sensitivity from both the channels become closer in value.

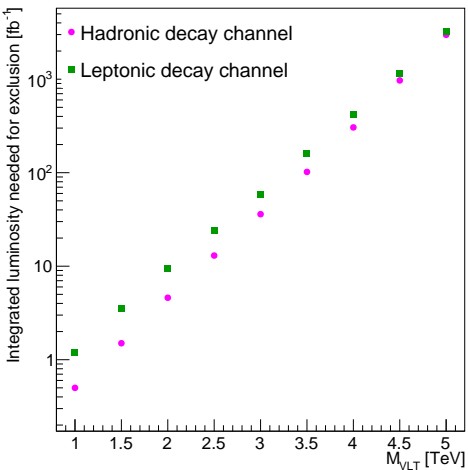

Figure 10: The minimum integrated luminosity needed to exclude VLT mass points are shown separately for hadronic and leptonic decay channels of the top quark, assuming we observe no events, but expect at least three signal events.

## 4 Conclusions

A maverick top partner model, decaying to a dark photon was suggested in Ref. [1]. A phenomenological exploration of the model has been performed, and a search strategy has been proposed exploiting the unique signal topology. We show that for a set of kinematic selections, both in hadronic and leptonic decay channels of the SM top quark, almost all background can be eliminated and the search is viable even with currently available LHC dataup to VLT mass of 3.5 TeV.

## Acknowledgements

We thank the authors of the maverick top partner paper, specifically Ian M. Lewis and Samuel D. Lane for providing us with model files to cross check ours. We thank Peter Loch for illuminating discussion on applicability of jet mass calibration and suggesting the use of mass over transverse momentum ratio. We thank Xifeng Ruan for discussion on statistical methods. DK is funded by National Research Foundation (NRF), South Africa through Competitive Programme for Rated Researchers (CPRR), Grant No: 118515. MMF is funded by the Department of Science and Technology (DOST) - Accelerated Science and Technology Human Resource Development Program (ASTHRDP) Postdoctoral Research Fellowship. SS thanks University of Witwatersrand Research Council for Sellschop grant and NRF for Extension Support Doctoral Scholarship. KDP thanks the National Astrophysics and Space Science Programme (NASSP) for their ongoing financial support. HVDS thanks SA-CERN programme for excellence scholarship.



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
