# Peer review of "Hitting two BSM particles with one lepton-jet: search for a top partner decaying to a dark photon, resulting in a lepton-jet"

_SciPost Physics, doi:SciPost Phys. 13, 018 (2022)_

## Round 2 · Referee Report · Anonymous (Referee 1) · 2022-3-24

Strengths

1) The authors study an interesting signal that would be interesting to search for.

2) The authors provide a thorough collider study of the signal, showing the viability of a search.

3) The analysis seems solid and reliable.

Weaknesses

1) The model is not well laid out in the paper, and there is some confusion in the discussion of what the decay of a top partner to a dark photon may be the dominant decay.

2) Overall, I found the discussion of the collider analysis to be a somewhat confusing. The authors could strengthen the paper by addressing many of the points in the report.

Report

This paper studies a search for novel decays of up-type vectorlike quark, i.e. top partners. In particular, the authors investigate the signal of a top partner decaying into a top quark and very light dark photon. The dark photon then decays into highly collimated leptons that form a lepton-jet. This is an interesting signal, in particular since top partners have not been discovered in more traditional modes. The authors have a complete collider study showing the viability of searching for this signal. The paper is publishable, once some concerns/questions are addressed and the discussion made clearer. The basis of a good paper is there and the material is substantive enough for SciPost, once the authors address the "Requested changes."

Requested changes

I would like to stress that although the "Requested changes" are quite long, the study appears reliable and trustworthy. I do not ask the authors to redo an analysis, just to clarify the following points, all of which should be addressed in the manuscript.

More substantive:

(1) In the third paragraph of page 3, it's stated that the top partner decay into dark photons is dominant when the dark photon is much less massive than the top partner. However, I think the requirement is that the dark sector vev is much smaller than the electroweak vev. As stated in the next paragraph, enhancement of the decay into the dark photon goes as (v_EW/v_d)^2. In principle, there could be a small dark photon mass by having a very small gauge coupling and large dark sector vev. In that case, there would be no enhancement to the top partner decay. It would be good if the authors could clarify the discussion.

(2) On the transition between pages 6 and 7, there is a statement "Here we have scaled the cross-sections by 0.1 to mimic the electron misidentification rate." Is there a relevant citation for this? I'm a bit surprised that the misidentification rate is so large, since I would expect most hadronic jets to leave deposits in the HCAL and the lepton jets to leave deposits in the ECAL only. Are these single pion backgrounds, or something similar? A clarification in the text would be useful.

(3) (a) In the first paragraph of page 4, the authors state that single production of very heavy vector like quarks dominates the QCD pair production. However, the single production is model-dependent and depends on the mixing of the vector like quark with the SM quarks. The QCD pair production is more model independent, depending on color representation, mass, and spin. Hence, if the mixing of the vectorlike quark with the SM top quark is very small, single production rates vanish. An illustration of this can be found in Fig. 3 of arXiv: 1803.06351 [hep-ph]. It would be good if the authors could clarify this in the manuscript. For example, the authors don't state what mixing angle they assume when calculating production rates.

(b) In section 2.2, the authors introduce a variable \varepsilon. This is not (and should be) defined in the paper.

(c) In section 2.2, the authors present some cross sections. In line with points (a,b) above, what parameters did the authors choose beyond the masses? How did they choose these parameters? These are very model-dependent cross sections.

(d) In general, the paper could benefit from having a section outlining the model, the quantum numbers, the interactions, the relevant parameters, and the current constraints on those parameters.

(4) On the bottom of page 8, the authors list that there are 2,380 remaining for vectorlike quark masses of 1 TeV, 35 events for 3 TeV, and 1 event for 5 TeV. Are these the number of events remaining from the original 100,000 generated for each signal mass? Or are these the number of events expected to remain after 300 inverse femtobarns of data? The number of events expected after 300 inverse femtobarns of data, or even 3 inverse attobarns, seems more useful since realistic significances could then be calculated.

(5) From tables 1 and 2, can the authors calculate the expected significance for their parameter point? This would seem like the useful information needed to determine if this signal is indeed observable at the LHC. The remaining cross section, and significances at 300 inverse femtobarns and 3 inverse attobarns would be very useful for the reader. Even a luminosity plot showing the luminosity for 2\sigma and 5\sigma observations would be helpful in understanding how observable this signal is.

(6) In section 3, I got a little lost in the discussion of jets. The word jet seems to simultaneously mean lepton and hadronic jets. However, these objects are very different. The entire discussion could be greatly improved by being more clear when the authors mean hadronic or lepton jets. This occurs throughout the discussion. A few questions along those lines:

(a) At the top of page 5, the authors mention using the mass over transverse momentum ratio to discriminate lepton-jets. However, can't you tell a lepton-jet from a hadronic jet mostly by the activity in the ECAL and the HCAL? Are the authors considering hadronic jets faking lepton-jets?

(b) In the first paragraph of Sec. 3.2, the authors say they require at least three jets. Is this one lepton jet and two hadronic jets?

(c) At the end of the second paragraph of section 3.2, the authors say they require one small radius jet and at least three large radius jets. However, I thought the signal only had one final state top quark, which would seem to be one large radius jet and not three? Or are they resolving the decay products of the top quark, which is different than what is shown in Fig. 2? I know the authors state that Fig. 2 is "not an accurate indicator of the event kinematics", but they do discuss top jets.

(d)In general, I thought the discussion of the collider analysis could be made clearer. I had similar confusion throughout the rest of the section.

Minor confusions:

(7) I was a little confused at the end of Sec. 2.2, where the authors state that there will be a combined 300 inverse femtobarn of data at the end of Run 3. Don't CMS and ATLAS already have a combined luminosity near 300 inverse femtobarns? Or do the authors means a combination of Run 1, 2, and 3 data?

(8) I may have missed it, but is Fig. 1 ever referred to in the text? If not, it should be.

(9) At the bottom of page 5, there is a statement "The decay mode of the top quark on the other side will determine the full topology." What is meant by "on the other side"?

Stylistic comments:

(10) Fig. 1 shows a diagram for single production of the vector-like quark and its decay. However, I think the t-channel production with a b-quark in the initial state is the dominant production mode.

(11) I think all the captions for figures and tables could be improved. In particular in Secs. 3 and 4 I found them confusing.

(12) In general, the paper is not well-cited. There are many earlier seminal studies of dark photons that have not been cited. The original papers studying lepton-jets are not cited, in particular the theory proposals of lepton-jets are not cited. Several of the signals discussed in Sec. 2.1 need citations. There are many citations missing throughout the paper.

(13) This comment the authors can take or leave. I thought the third paragraph of page 3 was was very motivation for this search and may be better placed in the introduction.

  • validity: top
  • significance: high
  • originality: high
  • clarity: ok
  • formatting: good
  • grammar: good

Author:  Deepak Kar  on 2022-05-10  [id 2452]

(in reply to Report 1 on 2022-03-24)

We thank the referee for carefully reading the paper, and making suggestions which definitely helped to improve the quality of the paper. Please find our responses below, which have been implemented in arXiv v3.

More substantive:

(1) In the third paragraph of page 3, it's stated that the top partner decay into dark photons is dominant when the dark photon is much less massive than the top partner. However, I think the requirement is that the dark sector vev is much smaller than the electroweak vev. As stated in the next paragraph, enhancement of the decay into the dark photon goes as (v_EW/v_d)^2. In principle, there could be a small dark photon mass by having a very small gauge coupling and large dark sector vev. In that case, there would be no enhancement to the top partner decay. It would be good if the authors could clarify the discussion.

We thank the referee for this observation. Indeed, as shown in Section I of Kim, et. al., Phys. Rev. D, 101, 035041 (2020), the enhancement goes as (v_EW/v_d)^2 and that VLQ preferentially decays to dark photons due to the mass gap between the dark sector bosons and SM bosons. Also, “[s]ince there is a quadratic dependence on v_EW and v_d, this mass gap does not have to be very large for the decays T -> t + gamma_d to be dominant”. We have now added a footnote in the paper clarifying that it is the vev requirement that causes the enhancement, but of course, a larger mass gap of m_gamma_d << m_Z only serves to amplify this enhancement further.

(2) On the transition between pages 6 and 7, there is a statement "Here we have scaled the cross-sections by 0.1 to mimic the electron misidentification rate." Is there a relevant citation for this? I'm a bit surprised that the misidentification rate is so large, since I would expect most hadronic jets to leave deposits in the HCAL and the lepton jets to leave deposits in the ECAL only. Are these single pion backgrounds, or something similar? A clarification in the text would be useful.

This is indeed an uber-conservative estimate, based on Table 3 of https://arxiv.org/abs/1612.01456. Even though the reference is a bit dated, and all the numbers are < 10%, they are also in a lower pT range, so we feel going with a slightly higher estimate for our high pT electrons should be ok. These are resulting from hadrons misidentified as electrons, electrons from semileptonic decay or photon conversions, and there are punch-through from em to hadronic calorimeters. We have now added this reference.

(3) (a) In the first paragraph of page 4, the authors state that single production of very heavy vector like quarks dominates the QCD pair production. However, the single production is model-dependent and depends on the mixing of the vector like quark with the SM quarks. The QCD pair production is more model independent, depending on color representation, mass, and spin. Hence, if the mixing of the vectorlike quark with the SM top quark is very small, single production rates vanish. An illustration of this can be found in Fig. 3 of arXiv: 1803.06351 [hep-ph]. It would be good if the authors could clarify this in the manuscript. For example, the authors don't state what mixing angle they assume when calculating production rates.

Indeed that’s a good point. We made it clearer in the manuscript now.

(b) In section 2.2, the authors introduce a variable \varepsilon. This is not (and should be) defined in the paper.

ε is the kinetic mixing parameter. (It is defined on page 4 just before Section 2.2

(c) In section 2.2, the authors present some cross sections. In line with points (a,b) above, what parameters did the authors choose beyond the masses? How did they choose these parameters? These are very model-dependent cross sections.

Details have now been added in Section 2.2 regarding the values of the free parameters. Again, the values were chosen in order to remain in the parameter space where VLTs are singly produced and that the dark photons decay entirely into electron lepton jets.

(d) In general, the paper could benefit from having a section outlining the model, the quantum numbers, the interactions, the relevant parameters, and the current constraints on those parameters.

We have added a table containing the relevant charges and field content of the new model as well as the new free parameters that arise. However, we still defer the entire discussion of the full model (with all the Lagrangians and interactions) to the original paper, otherwise we will only be repeating the derivations that they have already done.

(4) On the bottom of page 8, the authors list that there are 2,380 remaining for vectorlike quark masses of 1 TeV, 35 events for 3 TeV, and 1 event for 5 TeV. Are these the number of events remaining from the original 100,000 generated for each signal mass? Or are these the number of events expected to remain after 300 inverse femtobarns of data? The number of events expected after 300 inverse femtobarns of data, or even 3 inverse attobarns, seems more useful since realistic significances could then be calculated.

Indeed, we apologise that we did not make it clear that this corresponds to the 300 /fb. We have now made it clear, also please see next point about luminosity exclusion plot.

(5) From tables 1 and 2, can the authors calculate the expected significance for their parameter point? This would seem like the useful information needed to determine if this signal is indeed observable at the LHC. The remaining cross section, and significances at 300 inverse femtobarns and 3 inverse attobarns would be very useful for the reader. Even a luminosity plot showing the luminosity for 2\sigma and 5\sigma observations would be helpful in understanding how observable this signal is.

So this is essentially a zero background search, at least in particle level. So we cannot really calculate a significance number. However, we like the suggestion of quantifying the feasibility of the search as a function of integrated luminosity. We added: Although this is probably an idealised assumption in a particle level study, where detector effects and mis-measurements will result in a non-zero background, we can estimate the integrated luminosity needed as a function of VLT mass to exclude the signal if we observe no events, but expect at least three signal events. This is shown separately for hadronic and leptonic channels in Fig, where we can see Run 2 data may already be sufficient to have sensitivity upto VLT mass of 3.5~TeV.

(6) In section 3, I got a little lost in the discussion of jets. The word jet seems to simultaneously mean lepton and hadronic jets. However, these objects are very different. The entire discussion could be greatly improved by being more clear when the authors mean hadronic or lepton jets. This occurs throughout the discussion. A few questions along those lines:

Indeed the discussion became confusing with different kinds of jets being used interchangeably. We have now clarified by using J4 for antikt4 jets, J10 for anktikt10 jets, and specifying at the beginning that lepton-jet is a J4, and for hadronic channel, top-jet is a J10.

(a) At the top of page 5, the authors mention using the mass over transverse momentum ratio to discriminate lepton-jets. However, can't you tell a lepton-jet from a hadronic jet mostly by the activity in the ECAL and the HCAL? Are the authors considering hadronic jets faking lepton-jets?

This is a good point. A “clean” lepton jet will indeed deposit all its energy in ECal, but nothing is clean in a detector environment, so hadronic jets will definitely fake lepton jets. Unfortunately we cannot identify electrons and jets clearly based on energy fraction in ECal/Hcal, that is why we spend so much effort in designing complicated overlap removal methods. Therefore we have highlighted that m/pT is a good variable, but in a detector environment, energy fraction should also be looked at.

(b) In the first paragraph of Sec. 3.2, the authors say they require at least three jets. Is this one lepton jet and two hadronic jets?

We have clarified it now, it is indeed one leptonic jet candidate, and two hadronic jets.

(c) At the end of the second paragraph of section 3.2, the authors say they require one small radius jet and at least three large radius jets. However, I thought the signal only had one final state top quark, which would seem to be one large radius jet and not three? Or are they resolving the decay products of the top quark, which is different than what is shown in Fig. 2? I know the authors state that Fig. 2 is "not an accurate indicator of the event kinematics", but they do discuss top jets.

It was a typo, should have been the other way around, fixed now.

(d) In general, I thought the discussion of the collider analysis could be made clearer. I had similar confusion throughout the rest of the section.

We hope that it reads better now with the above-mentioned changes.

Minor confusions:

(7) I was a little confused at the end of Sec. 2.2, where the authors state that there will be a combined 300 inverse femtobarn of data at the end of Run 3. Don't CMS and ATLAS already have a combined luminosity near 300 inverse femtobarns? Or do the authors means a combination of Run 1, 2, and 3 data?

We now have 139 /fb from Run 2 at 13 TeV, and we typically do not use the 4.7 \fb from Run 1 as it was at 7+8 TeV. Our projected 300 \fb is combining Run 2 and Run 3. However we do note that Run 3 will be at 13.6 TeV, but that is still close enough to 13 TeV to “combine” the luminosity. We clarified this in the text now.

(8) I may have missed it, but is Fig. 1 ever referred to in the text? If not, it should be.

We have added it now, sorry about the omission before.

(9) At the bottom of page 5, there is a statement "The decay mode of the top quark on the other side will determine the full topology." What is meant by "on the other side"?

We were too casual here, it is probably sufficient to say the top quark, which we did now. We were trying to allude to the fact that top is almost always balanced by the lepton jet, but that is probably obvious anway.

Stylistic comments: (10) Fig. 1 shows a diagram for single production of the vector-like quark and its decay. However, I think the t-channel production with a b-quark in the initial state is the dominant production mode.

We have split the figure into production and decay modes now, including the t-channel production modes, which are indeed the dominant ones.

(11) I think all the captions for figures and tables could be improved. In particular in Secs. 3 and 4 I found them confusing.

We have updated the first two figures are their captions correspondingly. We have also added more details in the plot and table captions to help the reader to parse the figures better.

(12) In general, the paper is not well-cited. There are many earlier seminal studies of dark photons that have not been cited. The original papers studying lepton-jets are not cited, in particular the theory proposals of lepton-jets are not cited. Several of the signals discussed in Sec. 2.1 need citations. There are many citations missing throughout the paper.

We apologise for the oversight, a number of citations have been added now.

(13) This comment the authors can take or leave. I thought the third paragraph of page 3 was was very motivation for this search and may be better placed in the introduction.

We gave this a lot of consideration, but we felt it helps the flow better as it is now.

---

## Round 3 · Referee Report · Anonymous (Referee 1) · 2022-6-20

Strengths

The authors have completed a thorough collider study of an interesting signal.

Weaknesses

Minor typos/language.

Report

The authors have addressed all my concerns. The manuscript is much clearer, and the authors make a strong case that this is an interesting signal to look at.

After addressing my minor requested changes below, the paper is ready to be published.

Requested changes

  1. In the Introduction the authors state "This leads to the conjecture of the existence of a new gauge boson that would mediate this dark force, analogous to the role of the photon in Quantum Electrodynamics. For this reason, the new gauge boson is referred to in literature as the dark photon..." However, I believe the dark photon is referred to as a dark photon because its leading interactions in the small mixing limit is the QED current. There are other models with U(1) dark sector forces called "Dark-Zs" because they can interact with the weak neutral current as well. For example see arXiv:1203.2947.

  2. In the third paragraph on page 10 there's a statement "highest invariant pT" Do the authors mean just pT?

  3. In Fig. 6, the wording on before and after m/pT cuts and the placement of "(right)" are a little confusing.

  • validity: -
  • significance: -
  • originality: -
  • clarity: -
  • formatting: -
  • grammar: -

Author:  Deepak Kar  on 2022-06-22  [id 2600]

(in reply to Report 1 on 2022-06-20)

Dear referee,

Thank you for your follow up. We have included your suggestion for the introduction text, fixed the caption in Fig 6, and just used pair pT to avoid confusion in the new version.

Sincerely,
Deepak for the authors

---

## Round 3 · Author Response

We thanks the referee for their careful review of the paper, and we have addressed all the comments in this new draft. A detailed point by point response is posted as well.

---

## Round 3 · List of Changes

• Expanded the theory section with details of the model, parameter choices and references
  • Added a projected sensitivity study for the proposed search
  • Clarified the analysis section with specific labels for different types of jets used
  • Fixed typos, a few of which caused confusion

---

## Round 4 · List of Changes

We have included referee's suggestion for the introduction text, fixed the caption in Fig 6, and just used pair pT to avoid confusion in the new version.

---

## Editorial Decision

published